

# Phase space sampling and inference from weighted events with autoregressive flows

Bob Stienen[1*] and Rob Verheyen[2†]

**1** Institute for Mathematics, Astro- and Particle Physics IMAPP,
Radboud Universiteit, Nijmegen, The Netherlands
**2** Department of Physics and Astronomy, University College London,
London, United Kingdom

★ bstienen@science.ru.nl, † r.verheyen@ucl.ac.uk

## Abstract

We explore the use of autoregressive flows, a type of generative model with tractable likelihood, as a means of efficient generation of physical particle collider events. The usual maximum likelihood loss function is supplemented by an event weight, allowing for inference from event samples with variable, and even negative event weights. To illustrate the efficacy of the model, we perform experiments with leading-order top pair production events at an electron collider with importance sampling weights, and with next-to-leading-order top pair production events at the LHC that involve negative weights.



# 1   Introduction

With the increasing complexity of particle collision events at experiments like the LHC, the production of experimental predictions based on the Standard Model or other physical models has come to heavily rely on numerical simulations. General-purpose event generators like `Pythia`[1], `Herwig`[2] and `Sherpa`[3] are widely used Monte Carlo (MC) programs[4] that allow for direct comparison between theoretical predictions and experimental measurements.

As the amount of data gathered at the LHC increases, so does the required precision of the theoretical simulations. By now, the use of multiple-emission NLO generators through formalisms like MEPS@NLO[5,6] or FxFx[7] have become the standard. Furthermore, NLL accuracy in parton showers was recently achieved[8–11], and further improvements through the inclusion of higher-order splitting functions[12–14] and subleading colour effects[15–18] are now available. However, as a consequence the computational demands of MC event generation has sharply risen[19,20]. A significant component of the incurred computational cost of such simulations is due to the required computation of large-dimensional integrals that describe the phase space of LHC events. Monte Carlo techniques are often the only feasible option for these types of simulation, and as such efficient phase space sampling algorithms are required. While many commonly used techniques, like the `VEGAS` algorithm[21, 22], have been used successfully for a long time, their efficiency starts to suffer rapidly as the complexity of the sampling problem at hand increases[23], quickly becoming the bottleneck of the simulation pipeline. Many more traditional techniques have been proposed to improve performance[24–30], but the recent advances in the field of machine learning have lead to a number of highly promising algorithms which may be applicable to the problem of event generation in high energy physics more broadly.

In particular, Generative Adversarial Networks (GANs)[31], Variational Autoencoders (VAEs)[32] and several other architectures have been used successfully to sample events at various stages of the event generation sequence, and in many related high energy physics generative processes[33–64]. On the other hand, work done in[65–71] focused on the particular problem of sampling the phase space of a hard scattering process with different techniques, the latter making use of the relatively modern Normalizing Flow models[72]. While the use of GANs and VAEs has led to impressive results, they may in some cases be difficult to optimize. For example, the objective of a GAN is to find a Nash equilibrium between the generator and discriminator networks, which is generally difficult to find with gradient descent[73]. A VAE is instead trained to optimize a variational lower bound of the real likelihood, leaving leeway to mismodel the underlying probability distribution of the data. Flow models instead offer tractable evaluation of the likelihood, which may be optimized directly. This is a significant advantage in the context of the generation of high energy physics events, where precise reproduction of the density is paramount.

Normalizing flows are probabilistic models that are constructed as an invertible, parameterized variable transform starting from a simple prior distribution. Recent comprehensive reviews may be found in[74, 75]. The main challenge of the construction of these models is to ensure the variable transforms are both parametrically expressive and computationally efficient. To that end, much progress has recently been made[76–87]. In particular, the flow architecture first proposed in[76] along with the expressive transforms such as those proposed in[83, 88] were used in[68–70] as a phase space integrator. Autoregressive models[80, 81] generalize this architecture further by allowing for more flexible correlations between latent dimensions, and may as such be expected to perform better in larger feature spaces. This generalization comes at a higher computational cost in either the forward (sampling) direction or the backward (training) direction.

In this paper, we explore the use of autoregressive flows to sample the relatively high-

dimensional phase space of $t\bar{t}$-production and decay at the matrix element level in $e^+e^-$-collisions and at the parton shower level in $pp$-collisions at NLO. We show how flow models may be trained on event samples with variable and/or negative event weights. Such events occur frequently in the context of high energy physics, for instance in importance sampling for matrix element generation, the matching of higher order of perturbation theory to parton showers, or in the modelling of a combination of background and signal processes.

The two test cases explored in this paper are meant to demonstrate the ability of an autoregressive flow to be trained on weighted events, but also illustrates that it may be used as a phase space integrator as in [68–70], but also as a more general event generator like the other generative models of [33–64].

In section 2 we introduce the autoregressive flow architecture used in this paper. Section 3 then describes the application of the flow model to matrix element-level $e^+e^- \rightarrow t\bar{t}$ with the full post-decay phase space. The flow is trained on an unweighted event sample and compared with VEGAS, as well as on several sets of weighted events to exhibit the capability of the model to be trained on weighted data. In section 4 the flow model is applied to parton-level $pp \rightarrow t\bar{t}$ matched with MC@NLO [89]. This process is challenging due to the high-dimensional phase space and the appearance of negative weights. An outlook is given in section 5. Some supplementary material regarding phase space parameterizations used in sections 3 and 4 is given in appendix A.

## 2  Phase Space Sampling with Autoregressive Flows

Autoregressive flows are a class of normalising flows, which are a type of machine learning model that is able to directly infer the probability distribution of provided training data. This distribution is modelled by applying a series of parameterized, invertible transformations

$$z_{i+1} = f_i(z_i; \theta_i),\tag{1}$$

to a generally simple prior distribution $p_0(z_0)$, where $\theta_i$ are parameters that are determined during training. Applying the first transform leads to

$$
\begin{aligned}
p_1(z_1) &= p_0(z_0)\left|\det\frac{\partial f_1^{-1}(z_1;\theta_1)}{\partial z_0}\right|\\
&= p_0(z_0)\left|\det\left(\frac{\partial z_1}{\partial z_0}\right)^{-1}\right|\\
&= p_0(z_0)\left|\det\frac{\partial z_1}{\partial z_0}\right|^{-1},
\end{aligned}
\tag{2}
$$

where $\left|\det\frac{\partial z_1}{\partial z_0}\right|$ is the Jacobian determinant of the transform. The likelihoods $p_i(z_i)$ are often computed in log-space, yielding

$$\log p_1(z_1) = \log p_0(z_0) - \log\left|\det\frac{\partial z_1}{\partial z_0}\right|.\tag{3}$$

To model complex distributions, multiple transforms may be applied subsequently, leading to

$$\log p_K(x) = \log p_0(z_0) - \sum_{i=1}^{K}\log\left|\det\frac{\partial z_i}{\partial z_{i-1}}\right|,\tag{4}$$

where $x \equiv z_K$ are the real data features, and the other $z_i$ are latent variables. The training of the parameters $\theta$ of a normalising flow can be accomplished by minimising the negative log-likelihood

$$\text{Loss}(X, \theta) = -\sum_{x_i \in X} \log p_K(x_i) \tag{5}$$

of the training data $X$ under the modelled distribution using some form of gradient descent. As eq. (5) is a function of the parameters $\theta_i$ through the transforms $f_i(z_i; \theta_i)$, they may be iteratively updated toward a minimal negative log-likelihood through gradient descent.

Valid transformations $f_i$ need to be invertible and should have a Jacobian determinant that can be calculated efficiently, as this operation needs to be performed many times during inference. This last requirement becomes even more stringent as the dimensionality of the training data grows. The search for expressive transforms that adhere to these requirements has been one of the main paths in the research on normalising flows [72].

## 2.1 Autoregressive Flows

Autoregressive flows achieve the efficiency requirement by decomposing the likelihood for $D$-dimensional data such that it obeys the autoregressive property

$$p(z) = \prod_{j=1}^{D} p(z^j | z^{1:j-1}), \tag{6}$$

following the chain rule of probability, where superscripts indicate the feature of a data point (i.e. $z^2$ is the second feature of data point $z$). The likelihood is thus decomposed into a product of one-dimensional conditionals that may be modelled parametrically. In a flow model, eq. (6) may be imposed by casting the one-dimensional transformations in the form

$$z_{i+1}^j = f_{i+1}^j(z_i^j; \theta_i^j(z_{i+1}^{1:j-1})), \tag{7}$$

meaning that, to transform feature $j$ from $z_i^j$ to $z_{i+1}^j$, the parameters of the transform depend only on the previously calculated values for $z_{i+1}^1$ to $z_{i+1}^{j-1}$. The resulting Jacobian matrix is triangular and its determinant is the product of the diagonal entries, making the computation of its determinant very efficient. A normalizing flow that implements this idea is the Masked Autoregressive Flow (MAF) [80]. A visualisation of the procedure is shown in figure 1, where a diagram for both the forward-pass (required during sampling) and backward-pass (required during inference) is shown.

The parameters $\theta$ fed into the transformation function $f$ can be computed efficiently using a Masked Autoencoder for Distribution Estimation (MADE) network [79]. This is a deep neural network where internal connections are ignored such that the autoregressive property eq. (6) is satisfied. A diagrammatic illustration of a MADE network is shown in figure 2. Note that the choice to make $\theta$ depend on $z_{i+1}$ makes it impossible to parallelize the forward-pass through a MAF. The opposite is true for the backward-pass: as all values of $z_{i+1}$ are already known, this pass can be trivially parallelized.

The choice for a MAF architecture thus makes training, which boils down to backprop-agating the training data through the network and optimising the log-likelihood, relatively fast. Transforming data from the base distribution to the final distribution is, however, comparatively slow. One could alternatively choose to let $\theta$ depend on $z_i$, which would make the forward-pass fast, at the cost of a slower backward-pass. Such an architecture is known as an Inverse Autoregressive Flow (IAF) [81].

In either architecture the actual transformation function $f_i$ can be defined freely. They can be simple affine transformations [80], but a choice for more complicated functions can yield

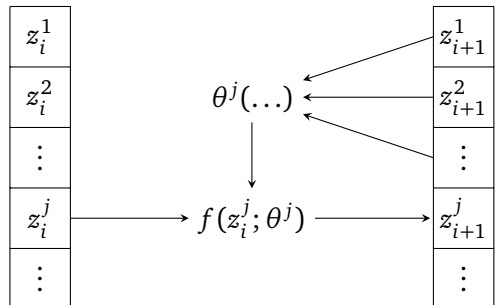

(a) Forward-pass in a Masked Autoregressive Flow.

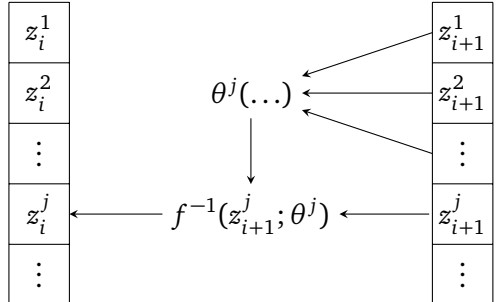

(b) Backward-pass in a Masked Autoregressive Flow.

Figure 1: Diagrammatic representation of a Masked Autoregressive Flow. The function $\theta^j$ takes as input the first $j-1$ entries of $z_{i+1}$. Its derivatives do not appear in the calculation of the Jacobian of this transformation, so $\theta^j$ can be implemented in the form of a neural network. The backward-pass can be paralellized, while this is not the case for the forward-pass.

a more expressive model. Furthermore, in this paper we explore the use of flow models in the sampling of phase space, which has well-defined boundaries. As such, it is convenient if the transforms are similarly restricted to a fixed domain. We therefore choose our transformations $f_i$ to be piecewise Rational Quadratic Splines (RQS), as defined in [83]. They are expressive, continuous and smooth $[0,1] \rightarrow [0,1]$ bijections that maintain easily calculable inverses and derivatives. The spline is spanned by a set of rational quadratic polynomials between a predetermined number of knots. The positions of the knots and the derivatives at those knots are parameterized by the MADE network in the form of bin widths $\theta_x^j$, bin heights $\theta_y^j$ and knot derivatives $\theta_d^j$. Figure 3 shows an illustration of a RQS.

## 2.2 Inference from Weighted Data

The autoregressive flow model explored in this paper may be trained on data from various stages of the event generation sequence, with the goal of either speeding up the process of event generation or adding statistical precision [90]. Often, traditional Monte Carlo techniques inevitably lead to the production of weighted events. Some examples are:

- Importance sampling of matrix elements with techniques such as VEGAS;

- Matching and merging of higher order calculations to parton showers;

- Scale variations in higher order calculations;

- Enhancement of rare branchings in parton shower algorithms;

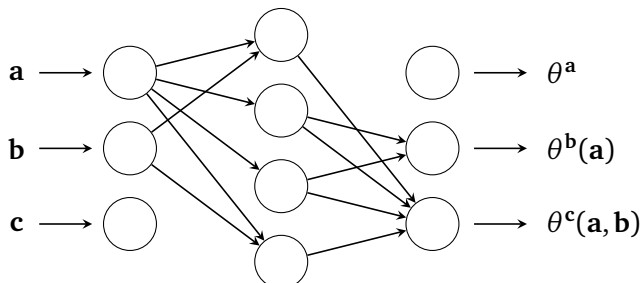

Figure 2: Graphical representation of a MADE network, which is a neural network in which specific weights have been masked, such that the autoregressive property of eq. (6) is obeyed. This figure shows the unmasked weights as arrows between the network nodes, which are indicated with circles.

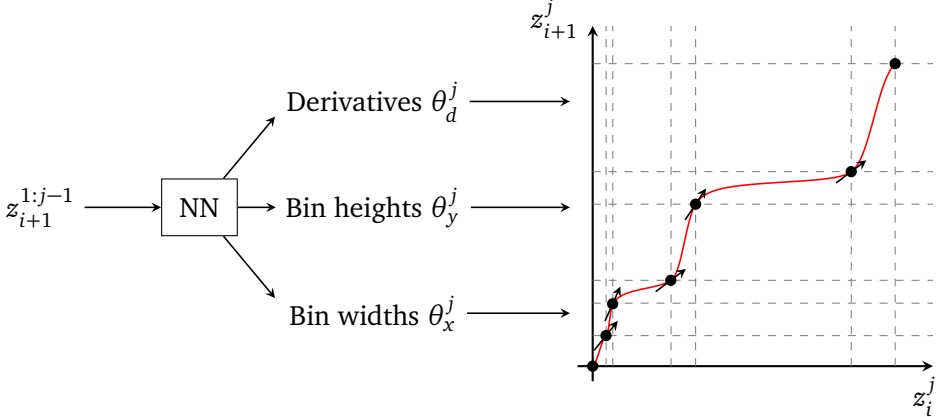

Figure 3: Visualisation of the construction of a rational quadratic spline $f$. A neural network takes parameters $z_{i+1}^{1:j-1}$ as input and returns the $y$-positions of the spline knots, the derivative at each of these knots and the distance between these knots. By spanning rational quadratic functions between these knots a monotonically increasing piecewise function that takes $z_i^j$ as input and that produces $z_{i+1}^j$ as output can be constructed.

- Enhancement of suppressed kinematic regions;

- The combination of event samples with strongly varying cross sections.

In some of the above cases, negative event weights may even appear.

Generally, weighted event samples may be unweighted through rejection sampling, where every event with weight $w_i$ is kept with probability

$$p_i = \frac{w_i}{w_{\max}}. \tag{8}$$

However, especially in cases where the event sampling procedure is computationally expensive and the fluctuation of weights is large, rejection sampling may be wasteful. Furthermore, it does not provide a solution to the appearance of negative weights, which are especially detrimental to statistical efficiency. However, recent work has shown that other methods are available to reduce the occurrence of negative weights [91–93].

Alternatively, the flow model can be trained on weighted events directly by a minor modi-

fication of the loss function of eq. (5):

$$\text{Loss}_{\text{weighted}}(X, \theta) = -\sum_{x_i \in X} w_i \log p_K(x_i). \tag{9}$$

This loss function leads to the correct optimization of the flow model even for event samples that include negative weights. Furthermore, it may be used to train the model more accurately in cases where it would be difficult to obtain sufficient statistics with unweighted events in suppressed corners of phase space. A flow model trained on weighted events will still generate unweighted events.

In the next sections, we perform experiments with weighted events obtained through importance sampling (section 3), and negatively weighted events due to matching to next-to-leading order (section 4).

### 2.3 Implementation

The flow model starts from a uniform base distribution and applies a number of RQS transforms, each with its own dedicated MADE network, between which the features are permuted to ensure full dependence of every feature on all others. The complete architecture is defined by the following hyperparameters:

- `n_RQS_knots`: the number of knots in every RQS;

- `n_made_layers`: the number of hidden layers in the MADE networks;

- `n_made_units`: the number of nodes in the hidden layers of the MADE networks;

- `n_flow_layers`: the number of RQS transformations applied to the base distribution.

To train the flow models, the Adam optimiser [94] is used with default settings. Additionally, a learning rate scheduling is applied: after a predefined number of epochs the learning rate is halved if the number of elapsed epochs is a multiple of a predefined period. The training of the flow models is defined by the following hyperparameters:

- `batch_size`: the number of data points in each training batch;

- `n_epochs`: the number of epochs for which the flow is trained;

- `adam_lr`: the initial learning rate of the Adam optimizer;

- `lr_schedule_start`: the epoch after which learning rate scheduling is started;

- `lr_schedule_period`: the number of epochs after which the learning rate is halved.

All experiments are performed using a modified version of `nflows 0.13` [95], which is built upon `PyTorch 1.6.0` [96]. The code and Jupyter Notebooks used in these experiments can be found in [97].

## 3 Experiments with Importance Sampling Weights

To test the performance of the autoregressive flow when trained on data with positive, fluctuating event weights in a particle physics context, the importance sampling of a matrix element is a natural candidate as it allows for straightforward definition of performance metrics. We thus consider the sampling of the phase space according to the LO matrix element of the process

$$e^+ e^- \to t\bar{t} \to (bud)(\bar{b}e^- \nu_e). \tag{10}$$

This process has been used as a benchmark in other work [37, 61], representing a challenging high-dimensional phase space that does not require any infrared cuts. After imposing momentum conservation and on-shell conditions, the remaining 14 dimensions are mapped to variables in the unit box $[0, 1]^{14}$ representing the top and $W$ resonance masses and solid angles in their respective decay frames. Further details may be found in appendix A.

We implement the VEGAS algorithm to compare the flow model with. The matrix element is retrieved from the C++ interface of MadGraph5_aMC@NLO [98]. The VEGAS algorithm is initialized from a prior distribution that mirrors the approximate Breit-Wigner shapes of the resonance masses using a mass-dependent width [99], and uniform distribution for all other dimensions[1]. The VEGAS results shown below represent performance after the integration grid is stable and the algorithm no longer improves.

To obtain events that are distributed according to the squared matrix element, one samples events $x$ from VEGAS or the autoregressive flow and assigning weights

$$w_i \propto \frac{|M(x)|^2}{p_{\text{sampler}}(x)},\tag{11}$$

where the $x$-independent proportionality factor may include constants associated with the phase space. Next, rejection sampling with acceptance probability (8) is applied, losing a fraction of the events, but ensuring the remaining sample follows the matrix element. To test the performance of VEGAS and the autoregressive flow model, we can compute the average unweighting efficiency

$$\eta = \frac{1}{n} \frac{\sum_{i=1}^{n} w_i}{w_{\text{max}}},\tag{12}$$

which indicates the average fraction of events left after rejection sampling. As was pointed out in [69, 71, 100], the straightforward definition is prone to outliers in the weight distribution. We follow [71] and clip the maximum weight to the largest Q-quantile of $w_i$, denoted by $w_Q$. The error in the Monte Carlo integral due to this clipping may be quantified by the coverage

$$\text{cov} = \frac{\sum_i w_i'}{\sum_i w_i},\tag{13}$$

where

$$w_i' = \begin{cases} w_i & \text{if } w_i \leq w_Q \\ w_Q & \text{if } w_i > w_Q. \end{cases}\tag{14}$$

One other often-used efficiency measure is the effective sample size [101, 102], which represents the approximate number of unweighted events that a weighted set would be equivalent to. We find that this measure is similarly sensitive to outliers, and thus restrict ourselves to the above-defined clipped unweighting efficiencies.

## 3.1 Unweighted Event Training

We first evaluate the inference capacity of the autoregressive model by training it on a set of $10^6$ event samples generated by VEGAS and unweighted through rejection sampling. Table 1 lists the values of the hyperparameters. Note that the method of training on pregenerated events differs from the approaches used in [68–71], where training is performed by sampling from the flow and evaluating the matrix element directly. While the architecture employed here is equally capable of this type of training, its parallelizable nature and the relatively large dimensionality of the process at hand means that training on a GPU is very beneficial. However,

---

[1]We find that VEGAS does not converge when initialized from a flat prior.

Table 1: Table of hyperparameters used for the importance sampling experiments.

| Model | | Training | |
|---|---|---|---|
| Parameter | Value | Parameter | Value |
| n_RQS_knots | 10 | batch_size | 1024 |
| n_made_layers | 1 | n_epochs | 800 |
| n_made_units | 100 | adam_lr | $10^{-3}$ |
| n_flow_layers | 6 | lr_schedule_start | 350 |
| | | lr_schedule_period | 75 |

to our knowledge there currently is no straightforward way to evaluate matrix elements on a GPU in the PyTorch framework, although progress has been made previously [103] and neural network-based approaches exist [104, 105].

Table 2 shows a comparison of the unweighting efficiencies of eq. (12) and the associated coverages of eq. (13) evaluated for the VEGAS algorithm and the flow model[2]. For reference, the efficiency of a flat sampling of the phase space parameterization is also included. The flow model outperforms VEGAS almost everywhere, with the notable exception of the coverage for $Q = 0.99999$. This indicates that, while the average unweighting efficiency is better, the flow model produces a few outliers with larger weights than VEGAS.

Figure 4 shows a number of distributions comparing the model with the Monte Carlo truth. That is, the VEGAS, flow and flat distributions represent the direct outputs of the samplers, while after weighting and performing rejection sampling, all samples will follow the true distribution. The top two panels show the $W$ boson mass and the $t$ quark azimuthal angles, which directly correspond with features of the space parameterization. Consequently, correlations between variables are not required and VEGAS and the flow model perform equally well. While the Breit-Wigner peak is completely absent in the flat distribution of the $W$ boson mass distribution, VEGAS and the autoregressive flow model it well, with VEGAS outperforming the flow slightly. However, the VEGAS algorithm has to be started from a Breit-Wigner prior distribution to achieve convergence, while the flow model is able to learn it without assistance. It is possible to select a different phase space parameterization that smooths out the Breit-Wigner peaks. In these experiments, the masses were purposely kept as features such that the capability of the flow model to learn such rapidly-changing distributions could be evaluated. The azimuthal distribution is included because the modelling of a flat distribution in the flow model is not necessarily any easier than any other shape.

The other four distributions shown are of the electron and $b$ quark energy, the $W$ boson transverse momentum and the angle between the $b$ and $\bar{b}$ quarks. These distributions are related to the phase space parameterization through a series of Lorentz transforms, meaning that correlations between dimensions are required to obtain the most accurate predictions. Consequently, VEGAS performs worse than the flow model across all spectra. The flow model predominantly mismodels the distributions in regions of low statistics. These types of discrepancies may in principle be remedied by biasing the training data towards the tails of distributions, and correcting in the event weights. We finally point out that the flat distribution is sampled in the hypercube phase space parameterization, and does thus not appear as flat after conversion to physical momenta.

---

[2] While the Masked Autoregressive Flow architecture is slower in the sampling direction than in the inference direction, the sampling of events on a GPU is still very fast. Sampling $10^6$ events takes approximately 24 seconds on a Nvidia GTX1080 Ti.

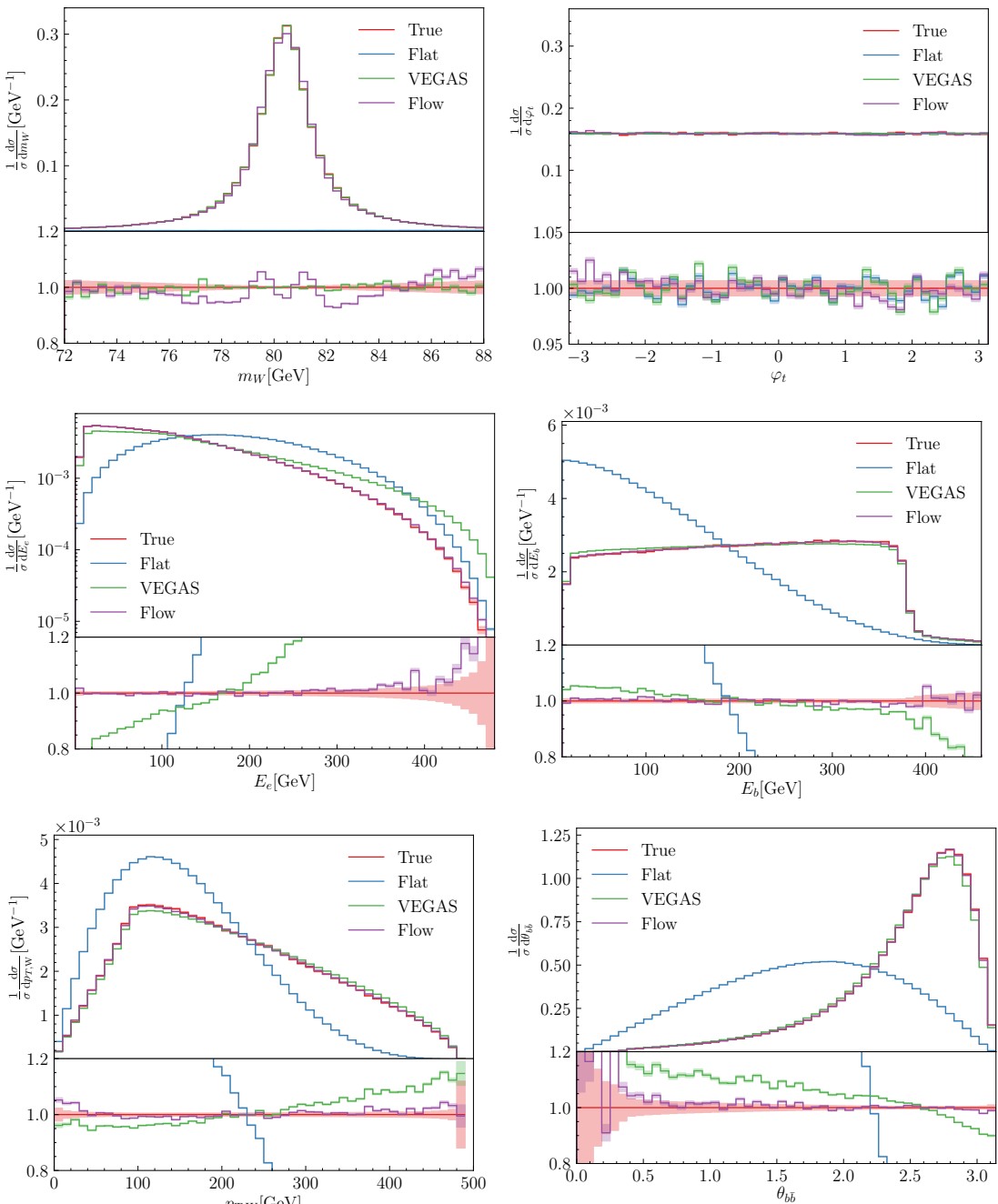

Figure 4: Distributions of the $W$ boson mass (top left), $t$ quark azimuthal angle (top right), electron energy (middle left), $b$ quark energy (middle right), $W$ boson $p_\perp$ (bottom left) and the angle between the $b$ and $\bar{b}$ quarks (bottom right) of the MC truth (red), which here serves as the training data, the VEGAS prediction (blue) and the flow model prediction (green).

Table 2: Table of unweighting efficiencies computed with $10^7$ flow, VEGAS and flat samples. The unweighting efficiencies and coverages are computed for three values of $Q$, where the first one corresponds with setting $w_Q = w_{\max}$.

|  | $Q = 1$ | | $Q = 0.99999$ | | $Q = 0.999$ | |
|---|---|---|---|---|---|---|
|  | $\eta$ | cov | $\eta$ | cov | $\eta$ | cov |
| Flow | 0.010 | 1 | 0.088 | 0.99987 | 0.32 | 0.9985 |
| VEGAS | 0.0082 | 1 | 0.077 | 0.99992 | 0.25 | 0.997 |
| Flat | $3.2 \cdot 10^{-7}$ | 1 | $2.6 \cdot 10^{-4}$ | 0.094 | $1.8 \cdot 10^{-3}$ | 0.0016 |

## 3.2 Weighted Event Training

In many cases, the bottleneck of importance sampling is not necessarily the likelihood evaluation, but rather the small unweighting efficiency achieved by commonly-used techniques [106]. We thus explore the capability of the normalizing flow to be trained on events generated by VEGAS before unweighting. We generate a sample of $10^6$ events with the same VEGAS setup used previously, and compute their importance sampling weights. We then consider the performance of the flow network when trained on the following three datasets:

**Weighted** The original $10^6$ data points with their importance weights;

**Unweighted** The remaining events after rejection sampling of the weighted data;

**Mean-weighted** Events are partially unweighted using the mean weight as a reference. That is, events that have weight $w < w_{\mathrm{mean}}$ are rejected with probability $w/w_{\mathrm{mean}}$ and assigned unit weight when kept, while events that have $w > w_{\mathrm{mean}}$ receive the adjusted weight $w/w_{\mathrm{mean}}$.

The left-hand side of figure 5 shows the distribution of weights of these datasets. The unweighted and mean-weighted sets retain respectively 0.83% and 70.78% of the original size.

The flow model is trained on the datasets above with the same hyperparameters as listed in table 1. The right-hand side of figure 5 shows the loss development during training. The unweighting efficiencies of eq. (12) and the associated coverages of eq. (13) are shown in table 3, and figure 6 shows the $W$ boson mass and electron energy distributions compared with the Monte Carlo truth. We observe very similar performance of the models trained on the weighted and mean-weighted datasets. However, the right-hand side of figure 5 shows that the latter converges faster. The slower convergence of the weighted dataset is a result of the large spread of weights, which causes large variance in the gradients during training and may lead to instability [107]. The model trained on the unweighted data fails to capture the Breit-Wigner peak and thus performs much worse. Too many events are indeed lost during rejection sampling, and figure 5 shows that the model would no longer improve upon further training.

## 4 Experiments with Negative Weights

To illustrate the capability of the autoregressive flow network to be trained on events with negative weights, we consider the process

$$pp \to t\bar{t}, \tag{15}$$

at next-to-leading order using MadGraph5_aMC@NLO, which interfaces with various external codes [108–113]. Events are generated at $\sqrt{s} = 13$ TeV with the NNPDF2.3 PDF sets [114],

Table 3: Table of unweighting efficiencies computed with $10^7$ samples drawn from the autoregressive flow models trained on the weighted, unweighted and mean-weighted datasets. The unweighting efficiencies and coverages are computed for three values of $Q$, where the first one corresponds with setting $w_Q = w_{\max}$.

|  | $Q = 1$ | | $Q = 0.99999$ | | $Q = 0.999$ | |
|---|---|---|---|---|---|---|
|  | $\eta$ | cov | $\eta$ | cov | $\eta$ | cov |
| Weighted | 0.00097 | 1 | 0.042 | 0.99954 | 0.27 | 0.9976 |
| Unweighted | 0.00040 | 1 | 0.010 | 0.9988 | 0.074 | 0.9911 |
| Mean-weighted | 0.0044 | 1 | 0.046 | 0.99977 | 0.26 | 0.9980 |

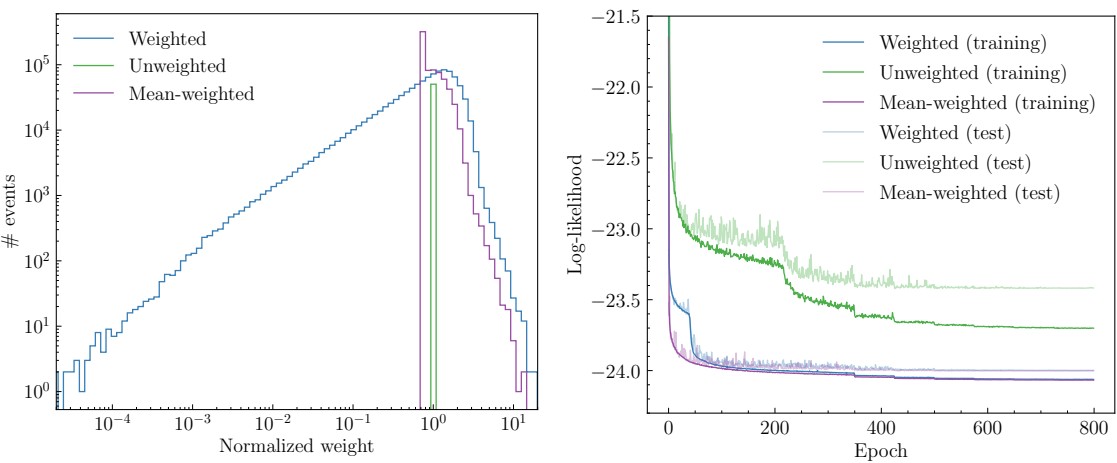

Figure 5: Distribution of weights of the weighted datasets, normalized to the size of the original weighted set (left) and the development of the training and test loss for the flow models trained on those datasets (right). The test loss is evaluated on the independent sample of the unweighted events used in section 3.1.

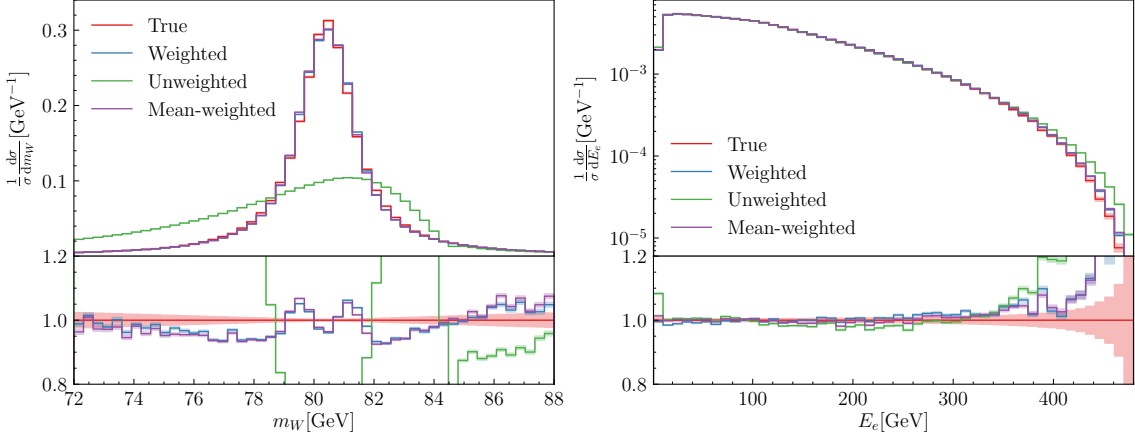

Figure 6: Distributions of the $W$ boson mass (left) and electron energy (right) of the MC truth (red) and the flow model trained on the weighted dataset (blue), un-weighted dataset (green) and mean-weighted dataset (purple).

Table 4: Table of hyperparameters used for the negative weights experiments.

| Model | | Training | |
|---|---|---|---|
| Parameter | Value | Parameter | Value |
| n_RQS_knots | 16 | batch_size | 1024 |
| n_made_layers | 3 | n_epochs | 800 |
| n_made_units | 250 | adam_lr | $10^{-3}$ |
| n_flow_layers | 12 | lr_schedule_start | 350 |
| | | lr_schedule_period | 75 |

decayed to $b$ quarks and leptons by `Madspin` [115] and are matched to the `Pythia8` parton shower through the MC@NLO prescription [89], which is a necessity to obtain physically sensible events. The parton-level events are clustered with the $k_t$ algorithm with $R = 0.4$ using `FastJet` [116]. The top quark momenta are then reconstructed as described in appendix A. We note that the flow could also be trained on hadron-level or detector-level events.

By default, `MadGraph5_aMC@NLO` produces events of which a fraction of 23.9% has a negative weight. Consequently, the dataset consists of $3.68 \times 10^6$ events, which would statistically correspond with $10^6$ unweighted events. The hyperparameter settings of the autoregressive flow are shown in table 4.

Figure 7 shows several distributions comparing the MC truth, the equivalent distribution ignoring the sign of the event weights, and the flow model prediction. The upper distributions are features present in the data directly, while the lower are observables that require correlations. The distributions without event weights are included here to show that the model indeed learns to incorporate negatively weighted events correctly during training. The effects of the negative weights are especially relevant in distributions like the transverse momentum of the top pair, which are determined by higher-order QCD corrections. The autoregressive flow matches the true distributions well, again only mismodelling some regions with low statistics.

# 5 Conclusion and Outlook

We have shown that autoregressive flows are a class of generative models that are especially useful in the task of sampling complex phase spaces. Unlike other generative models like GANs and VAEs, they have the distinct advantage of direct access to the model likelihood during inference, such that the loss function may be defined to directly fit the model density to the data density. Not only does this offer a clear objective during training, but it may also be very useful for other purposes such as the evaluation of the generalizability of the model, or for methods such as likelihood-free inference [117]. Furthermore, we show that the loss function can be generalized trivially to incorporate fluctuating and/or negative event weights.

We performed experiments with leading-order matrix element-level $e^+e^- \to t\bar{t}$ events with full decays. When trained on unweighted events, the autoregressive flow outperforms `VEGAS` and is able to learn the resonance poles automatically. The same model architecture was then trained on events with variable importance weights, and it was shown that better performance is achieved when compared with the equivalent unweighted dataset. This indicates that it may be beneficial to train on sets of weighted events when event generation is expensive or the unweighting efficiency is low. Such event sets commonly appear in the context of high energy physics, and weights may alternatively be used to improve model precision in regions of low statistics.

To show how the autoregressive flow deals with negative weights, it was trained on next-to-leading-order parton shower-level $pp \to t\bar{t}$ events. By inspecting observables that are sensitive

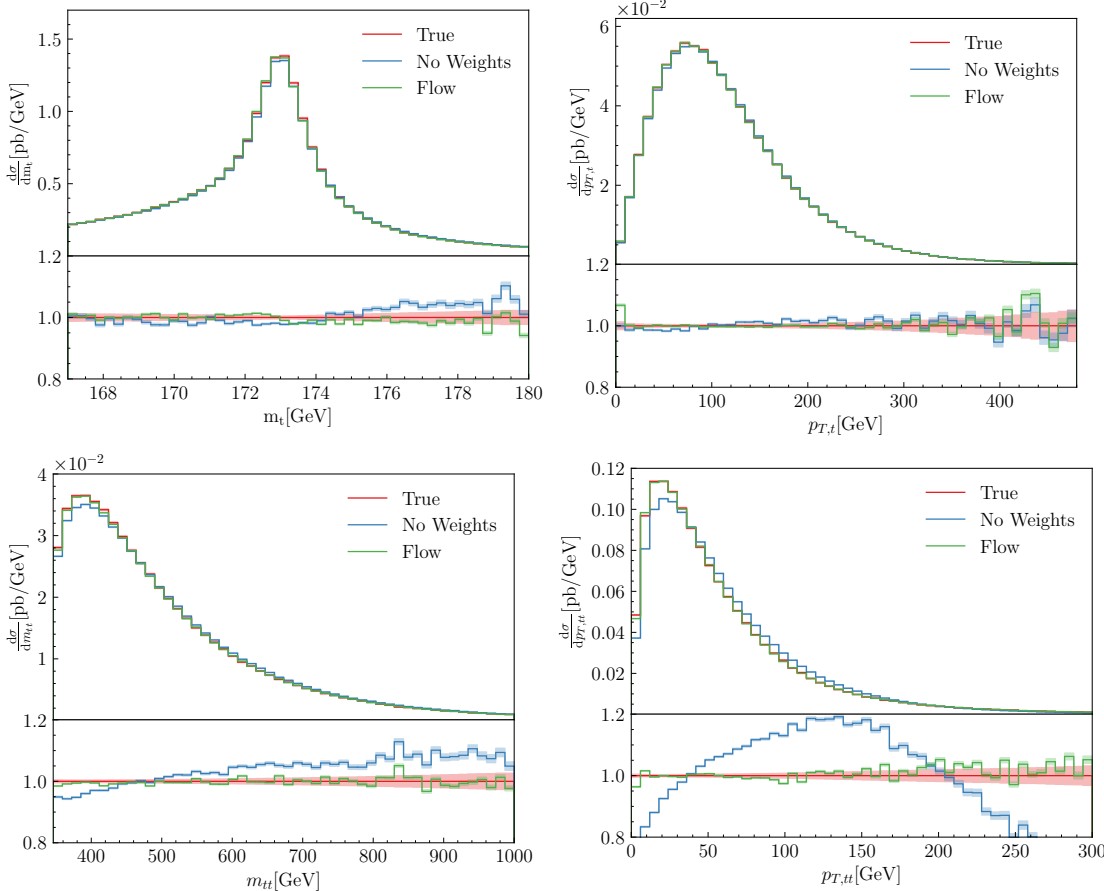

Figure 7: Distributions of the top mass (top left), the transverse momentum of the top pair (top right), the invariant mass of the top pair (bottom left) and the transverse momentum of the top (bottom right) of the MC truth (red), the MC truth ignoring the sign of the event weights (blue) and the flow model prediction (green).

to the higher-order QCD corrections to that process, it is made clear that the autoregressive flow is able to incorporate the negative event weights correctly.

The results presented here represent first evidence for the use of autoregressive flows as a potential alternative for traditional Monte Carlo techniques. Although there are differences between the true and modelled distributions, these are generally at the percent-level, except for regions with low statistics. However, as with almost any machine learning method, flow models are expected to exhibit improved performance when trained on more data, as long as adequate hyperparameters are chosen. Furthermore, the ability to perform inference from weighted data enables one to cover regions of low statistics more comprehensively, as long as the density is corrected through the event weights.

However, there is still a long ways to go before autoregressive flows, and generative models more generally, may function as a stand-in for a full-fledged event generator. While this would be highly beneficial in aspects like computational efficiency, better control over the systematic errors learned by the model is required. In this context, flow models may have an advantage over other options due to their directly tractable likelihood.

Furthermore, while the model presented here may be applied to any well-defined phase space with a fixed number of momenta, further improvements are required for the simulation of realistic final states that could even be inferred from data directly. For example, the number of objects is typically not fixed, which is not straightforwardly dealt with in the nor-

malizing flow paradigm. Furthermore, discrete features appear both in realistic events in the form of object labels, as well as at the matrix element level in the form of helicity and colour configurations, which are not straightforwardly accurately modelled by a continuous flow.

Finally, multiple types of generative model architectures exist, of which normalizing flows are the youngest and are still rapidly developing. While all of these models have been shown to be able to produce particle physics events efficiently and accurately, a thorough and systematic assessment of their accuracy is required before they may be considered as a stand-in for current MC event generators.

# Acknowledgements

We are grateful to Sascha Caron and Luc Hendriks for many useful discussions. RV thanks Stefan Prestel for help with the generation of the next-to-leading-order data. RV acknowledges support by the Foundation for Fundamental Research of Matter (FOM) via program 156 Higgs as Probe and Portal, by the Science and Technology Facilities Council (STFC) via grant award ST/P000274/1 and by the European Research Council (ERC) under the European Union's Horizon 2020 research and innovation programme (grant agreement No. 788223, PanScales). BS acknowledges the support by the Netherlands eScience Center under the project iDark: The intelligent Dark Matter Survey.

# A   Phase space parameterizations

In this appendix we briefly summarize the phase space parameterizations used in experiments of this paper.

## A.1   Leading Order $e^+e^- \rightarrow t\bar{t}$

The general phase space integration element is given by

$$d\Phi_n = (2\pi)^{4-3n}\delta\left(P - \sum_{i=1}^{n} p_i\right)\prod_{j=1}^{n} d^4 p_j\,\delta(p_j^2 - m_j^2), \tag{16}$$

where $P$ is the center-of-mass momentum which has $P^2 \equiv s$. Due to the appearance of multiple Breit-Wigner-like peaks in the process at hand, it is sensible to decompose the phase space into two-body elements connected by integrals over the invariant masses that appear in the amplitude propagators. In particular, we may write [118]

$$\begin{aligned}
d\Phi_6(p_b, p_{\bar{b}}, p_e, p_\nu, p_u, p_d | P) = {} & dm_{w^+}^2\, dm_{w^-}^2\, dm_t^2\, dm_{\bar{t}}^2 d\Phi_2(p_t, p_{\bar{t}} | P) \\
& \times d\Phi_2(p_{w^+}, p_b | p_t)\, d\Phi_2(p_{w^-}, p_{\bar{b}} | p_{\bar{t}}) \\
& \times d\Phi_2(p_u, p_d | p_{w^+})\, d\Phi_2(p_e, p_\nu | p_{w^-}),
\end{aligned} \tag{17}$$

where

$$\begin{aligned}
p_{w^+} &= p_u + p_d & p_t &= p_b + p_{w^+} \\
p_{w^-} &= p_e + p_\nu & p_{\bar{t}} &= p_{\bar{b}} + p_{w^-}
\end{aligned} \tag{18}$$

are the momenta of the intermediate resonances. The two-body phase space element may be written as

$$d\Phi_2(p_1, p_2 | q) = \frac{1}{32\pi^2}\lambda\left(1, \frac{p_1^2}{q^2}, \frac{p_2^2}{q^2}\right)^{1/2} d\Omega, \tag{19}$$

where $\lambda$ is the Källén function and $d\Omega \equiv d\cos(\theta)\,d\varphi$ is the solid angle integration element defined in the rest frame of $q$. Eq. (17) is then easily converted to an integral over the unit box $[0,1]^{14}$ by rescaling all masses and angles to

$$x_\theta = \frac{1}{2}(\cos(\theta)+1) \quad x_\varphi = \frac{\varphi}{2\pi} \quad x_{m^2} = \frac{m^2}{s}. \tag{20}$$

Transforming back and forth between the phase space and the unit box parameterization thus involves a number of Lorentz boosts between the rest frames of the intermediate resonances.

## A.2 Next-to-Leading Order $pp \to t\bar{t}$ with Parton Shower

The top quark decays are fixed to

$$\begin{aligned} t &\to b\,W^+ \to b\,\mu^+\,\nu_\mu \\ \bar{t} &\to \bar{b}\,W^- \to \bar{b}\,e^-\,\bar{\nu}_e. \end{aligned} \tag{21}$$

After parton showering and jet clustering, the momenta of the resonances are constructed as

$$\begin{aligned} p_t &= p_{j_b} + p_\mu + p_{\nu_\mu} \\ p_{\bar{t}} &= p_{j_{\bar{b}}} + p_e + p_{\bar{\nu}_e}, \end{aligned} \tag{22}$$

where $p_{j_b}$ and $p_{j_{\bar{b}}}$ are the momenta of the b-tagged jets. The top quark momenta are converted to the unit-box variables

$$x_m = \frac{m}{\sqrt{s}}, \quad x_{p_T} = \frac{p_T}{\sqrt{s}}, \quad x_\theta = \frac{1}{2}(\cos(\theta)+1), \quad x_\varphi = \frac{\varphi}{2\pi}, \tag{23}$$

where $m$ is the resonance mass, $p_T$ is the transverse momentum with respect to the beam direction and $\theta$ and $\varphi$ are the polar and azimuthal angles.

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
