# Peer review of "Phase Space Sampling and Inference from Weighted Events with Autoregressive Flows"

_SciPost Physics, doi:SciPost Phys. 10, 038 (2021)_

## Round 1 · Referee Report · Anonymous · 2021-1-6

Strengths

The presented work contributes to the current topic of employing machine learning techniques - here in particular autoregressive flows - to sample the phase space of particle collision final states. The authors study the capabilities of the flow method to infer the target distribution of top-quark production processes from pre-generated event samples.

1- The paper nicely introduces the subject and provides extensive reference to the relevant literature.

2- The authors employ cutting edge machine learning technology, i.e. the autoregressive flow method, to address the problem of event generation in high energy physics.

3- They consider relevant and non-trivial examples to test and benchmark their approach.

4- The paper is carefully and clearly written.

Weaknesses

To my understanding the authors should more clearly describe the scope of their approach and the differences with regards to Refs. [66-68]. In contrast to the work presented there, by inferring the target distribution from a limited set of pre-generated events, their generator does not necessarily provide events distributed according to the correct target distribution. The discussion of the results of their experiments in Secs. 3 and 4 misses this perspective, i.e. is somewhat misleading.

Report

The paper describes a novel approach to employ normalising flows as a generative model for sampling the phase space of HEP collision events. The authors consider the autoregressive flow method to infer the phase space distribution from pre-generated training sets of collision events. They use recent, state-of-the-art machine learning technology that promises significant potential to solve a pressing problem in contemporary high energy physics, namely the efficient sampling of complex, multi-dimensional phase spaces.

The proposed technique is applied to top quark pair production at leading and next-to-leading order, thereby confronting the ansatz with varying event weights, that in the latter case can be negative.

The authors convincingly show that their method captures broad features of the underlying distribution from the training samples provided, thereby in parts outperforming the well-known VEGAS sampler.

My general criticism concerns the classification of the approach proposed as a potential alternative for traditional Monte Carlo techniques. Even with the flow method providing bijective phase space maps, in their setup the sample is not guaranteed to produce events that follow the correct target distribution, as it is based on its inference from a limited set of training events. Their generative model, as apparent from Figs. 4 and 5 nicely approximates the true distribution but in certain regions of phase space produces significant differences.

The authors should more clearly specify and limit the scope of their approach and more carefully separate it from the work presented in Refs. [66-68]. This concerns in particular a more detailed discussion and interpretation of the found results, but also the claims made in the introduction and conclusions.

Requested changes

Given my comments from above, I would like to urge the authors to adjust the introduction, results discussion and conclusions accordingly. In particular:

1- Explain more clearly what is in fact shown in Figs. 4 & 5, if these are differential cross sections, they would be 'wrongly' predicted by the studied samplers. This is not compatible with the claim made in the conclusions: "first evidence for the use of autoregressive flows as a potential alternative for traditional Monte Carlo techniques".

2- How could the remaining deficiencies of the sampler systematically be cured?

3- For the flow and Vegas approaches, do the authors expect/see an improvement with an increase in training data statistics?

---

## Round 1 · Referee Report · Tilman Plehn · 2021-1-14

Strengths

1- the paper tackles some very serious challenges in LHC simulations
2- it employs new concepts and ideas
3- it represents the technical state of the art
4- the new ideas can easily be used by the Monte Carlo community

Weaknesses

1- some room for improvement in the presentation, as discussed below

Report

The paper is excellent, timely, technically state of the art, and should definitely be published in a leading physics journal.

Requested changes

Just front to back...
1- p.2, the dimensionality of phase space is less of a problem for VEGAS than for almost any other numerical tool, and I am not sure if phase space is really the leading bottleneck;
2- in the discussion of NFs as generative networks, I think our GAN and INN unfolding papers provide a useful benchmark in terms of mathematical foundation and performance;
3- p.6 I am not sure I got that right, is it true that their autoregressive flow is much more easily trained in one than in the other direction?
4- also p.6, I am sorry, but I find the discussion of the splines not very clear. What role do they play exactly?
5- somewhere at the end of Sec. 2.2 I would say clearly that the output of the flow network is a set of unweighted events, or not?
6- on p.10 it is not 100% clear what the different samplings mean. Flat is just a grid on the unit cube on one side of the NF, I assume? Where does the grid for the VEGAS sampling come from?
7- Sorry, but it is not clear to me how you compute the efficiencies in Tab.2 from the generated events, I am a little confused...
8- Fig.4, Any chance you could include the training data in those plots? I assume the True is before unweighting the training data? And I find the flat case a little useless, given that we have mass peaks which we know we cannot map out;
9- also in Fig.4, the y-range of the secondary panels is good only for one of the six panels;
10- in Fig.5, LHS, how are the curves normalized? Maybe include the integral in the caption to get a feel for things?
11- Fig.6, it would be nice to also see the training data. Why is the green mW peak so smooth and so off? That worries me in terms of error control. In the RHS it might be nice to also show a log scale, so the deviations in the tail are easier to spot? In the caption there is a typo green <-> blue;
12- Not sure if it is useful, but I was wondering how the trained network looks on the latent space side;
13- is ignoring the event weight really a good reference? I think what people sometimes do in real life is ignore the negative events and hold their breath;

---

## Round 2 · Referee Report · Tilman Plehn (Referee 2) · 2021-1-21

Report

Thanks for having a careful look! Just to clarify the one point, I had wondered how the latent space distributions looked in the different dimensions after training, just out of curiosity. But it's not really important, so here we go, very nice paper!

---

## Round 2 · Referee Report · Anonymous (Referee 1) · 2021-1-29

Report

I'd like to thank the authors for the careful revision of their manuscript. I am happy with all the done adjustments and clarifications and recommend the article for publication in its present form.

---

## Round 2 · Author Response

Dear editor & reviewers,

We are grateful to the reviewers for their careful reading of the manuscript and their detailed comments. Below, we address all of them individually.

Referee 1 (Anonymous)

1 - As part of the other referee’s comments, we have elaborated further on the exact meaning of the distributions in figures 4 and 5. Regarding the claim in the conclusions, we agree that this was not explained properly. We have included a more thorough explanation in the introduction and the conclusion, but in summary, models like the one used here can be used as in refs [66-68] to generate an exact distribution through weighting and rejection sampling, or one may envision using them as a stand-in for, or addition to, an event generator. For the second option to be viable, more precise control will be required over the systematic errors learned by the flow. We believe that, while the current state of the art has not accomplished that yet, normalizing flow-type models may be better suited for such a purpose compared with other generative models, because of its directly tractable likelihood.

2 - As pointed out below, the flow predominantly mismodels regions of low statistics. These may be cured by either using more training data, or cleverly adjusting the data such that such regions of low statistics are covered more comprehensively, and adjusting the event weights to retain the same distribution. We have clarified this further in the conclusion. Some deficiencies remain outside regions of low statistics, in particular in the W peak in figs. 4 and 5. Due to the rapidly changing value of the cross section around the peak, the flow model struggles to learn it as accurately as most other observables. In our experiments, we purposely kept the masses as features of the phase space to see how well the flow would be able to model them. However, with this prior knowledge, one might select a different phase space parameterization to smooth out the peak, and ease the training of the flow. We have added the above explanation to section 3.1 as well.

3 - As normalising flows are machine learning models, they get their information from training data. A general rule within machine learning is that the performance of models increases as more training data is used. We expect this also to be the case here. The fact that, in our experiments, most of the mismodelling of the flow is found in regions with little training data, strengthens this expectation. We have extended the conclusion of the paper to explicitly include this expectation.

Referee 2 (Mr. Plehn)

1 - We agree, and have slightly adjusted the wording in the introduction accordingly.

2 - We agree that this work should have been cited, and have included both papers now.

3 - The text explaining the parallelizable direction of a normalising flow was a bit too condensed, causing this misunderstanding. Depending on the type of flow layers, one of the directions can be parallelized, while the other cannot. In our case, we chose to ensure the inference direction is fast, such that training is faster. We have extended the text with a more elaborate explanation of this fact.

4 - We agree that this could have been made more clear in the previous manuscript. The splines are used as function f in the MAF layers (see Figure 1 in the manuscript). We added further clarification.

5 - This suggestion is indeed correct and we agree it is valuable to point out. We have extended Section 2.2 accordingly.

6 - We have tried to clarify the meanings of the different sampling at several points in section 3. In particular, the third paragraph of 3.1 now includes an additional sentence detailing the meaning the distributions shown in fig. 4. The flat distribution is earlier clarified to mean a flat sampling of the hypercube phase space representation detailed in the appendix. A further note explaining this has been added to the end of section 3.1.

7 - The definition of the event weights were indeed missing from the importance sampling section. They have now been added, and we hope this clarifies the meanings of the efficiencies.

8 - The model is trained on VEGAS events which have been weighted and rejection-sampled to follow the physical distribution. As such, True is the training data. This was already clarified as part of the previous point, we hope this is sufficiently clear now. We have also clarified this in the caption of figure 4. We believe that keeping the flat case in the plots is still sensible, as it is also referenced in table 2, and it puts the ability of VEGAS and the flow to get very close to the real distribution in perspective.

9 - We have zoomed in on the ratio plot of phi-b, because it is completely flat, but we prefer to maintain consistency among the other plots for better readability. It is difficult to find a common scale, because VEGAS deviates significantly in some regions, but we also want to show how close the flow model is able to get in others. We settled on 20% as it captures most regions with sufficient detail.

10 - The normalisation was chosen such that the blue ‘weighted’ histogram had an area under the curve of 1. However, when reading this question, we came to the conclusion that it would be more informative to not normalize at all and put the actual number of events on the y-axis. This change should clear any possible confusion about normalisation factors. The figure in the paper has been updated.

11 - As mentioned in 8, True is the training data (when the event weights are included), and the case where the event weights are ignored is shown as VEGAS in fig. 4. The model trained on unweighted data performs much worse than the others due to its small size. This is why the green curve misses the real distribution by a long shot. Hence, we advocate that, instead of unweighting the data, one may be better off training on weighted data directly. We have transformed to a log-scale for the right-hand side figure, and the corresponding panel in figure 4, but have kept the other panels as they were. The typo was fixed.

12 - We are not entirely sure what is meant by this comment. Assuming z_0 is meant as the latent space, the flow is trained to transform the training data to a uniform distribution. If the training is not perfect, the distribution may not be entirely uniform, but it may be difficult to draw any definite conclusions from that fact.

13 - Ignoring the event weights is indeed certainly not a good reference from a physical perspective. However, the corresponding distribution is included here to show that the network indeed incorporates the negative weights during training. We have elaborated on this a little more at the end of section 4.

---

## Round 2 · List of Changes

• Improved explanation on the difference between MAF an IAF layers in Section 2.1;
  • Improved explanation on the use of splines as transformation function in MAF layers (Section 2.1);
  • Added to the end of Section 2.2 explicitly that the flow outputs unweighted events;
  • Improved explanation on the origin of the weights w_i (Section 3);
  • Improved explanation of the data shown in Figure 4 in Section 3.1;
  • Added an explanation on the explicit incorporation of the W-boson Breit-Wigner peak in the features to be learned by the flow model (Section 3.1);
  • Added an explicit explanation on why the flat samplings are not flat in Figure 4.
  • Changed the LHS of Figure 5 to not show normalised distributions (with respect to the blue curve), but rather the absolute data counts instead;
  • Added an explanation (at the end of Section 4) on why we included the distributions without event weights in Figure 7;
  • Added possibilities for obtaining better results to the Conclusion.
  • Made more clear (in the Conclusion) that flows are not yet in a state that they can be used as stand-in for a full-fledged event generator;
  • RHS of Figure 6 and third panel of Figure 4 now have a logarithmic y-axis;
  • The second panel of Figure 4 has not a zoomed-in ratio plot.

---

## Editorial Decision

published